# Role of Mesenchymal Stem/Stromal Cells in Modulating Ischemia/Reperfusion Injury: Current State of the Art and Future Perspectives

**DOI:** 10.3390/biomedicines11030689

**Published:** 2023-02-23

**Authors:** Vitale Miceli, Matteo Bulati, Alessia Gallo, Gioacchin Iannolo, Rosalia Busà, Pier Giulio Conaldi, Giovanni Zito

**Affiliations:** Research Department, IRCSS ISMETT (Istituto Mediterraneo per I Trapianti e Terapie ad Alta Specializzazione), 90127 Palermo, Italy

**Keywords:** ischemia/reperfusion injury, inflammation, apoptosis, organ transplantation, mesenchymal stem/stromal cells, secretome, MSC pre-conditioning

## Abstract

Ischemia/reperfusion injury (IRI) is a multistep damage that occurs in several tissues when a blood flow interruption is inevitable, such as during organ surgery or transplantation. It is responsible for cell death and tissue dysfunction, thus leading, in the case of transplantation, to organ rejection. IRI takes place during reperfusion, i.e., when blood flow is restored, by activating inflammation and reactive oxygen species (ROS) production, causing mitochondrial damage and apoptosis of parenchymal cells. Unfortunately, none of the therapies currently in use are definitive, prompting the need for new therapeutic approaches. Scientific evidence has proven that mesenchymal stem/stromal cells (MSCs) can reduce inflammation and ROS, prompting this cellular therapy to also be investigated for treatment of IRI. Moreover, it has been shown that MSC therapeutic effects were mediated in part by their secretome, which appears to be involved in immune regulation and tissue repair. For these reasons, mediated MSC paracrine function might be key for injury amelioration upon IRI damage. In this review, we highlight the scientific literature on the potential beneficial use of MSCs and their products for improving IRI outcomes in different tissues/organs, focusing in particular on the paracrine effects mediated by MSCs, and on the molecular mechanisms behind these effects.

## 1. Introduction

Ischemia/reperfusion injury (IRI) is defined as organ damage due to cellular dysfunction. It usually occurs after organ surgery, and in particular during organ transplantation [1,2,3,4]. IRI is considered a multistep damage: for example, in transplantation, the first step is induced by organ resection and the inevitable ischemia, due to the interruption of blood flow and cold preservation of the organ for its storage before the transplant. During the ischemic phase, which varies according to the quality of the donor organ, disruption of blood flow and organ preservation at cold temperatures lead to mitochondrial dysfunction, intracellular adenosine triphosphate (ATP) depletion, and a switch to anaerobic metabolism [3]. Restoration of the blood flow during reperfusion exacerbates the damage, causing destruction of the function and viability of the organ. In particular, re-establishment of blood supplies causes an extensive production of pro-inflammatory cytokines and reactive oxygen species (ROS), which eventually leads to neutrophil infiltration and apoptosis of parenchymal cells [5,6]. IRI occurs in a wide range of organs, including lung, pancreas, kidney, gut, heart, brain, and liver, and can also induce systemic damage, potentially leading to multisystem organ failure [7]. Despite the variety of tissues involved, the biological processes activated by IRI are relatively common, and include inflammation, ROS production, and apoptosis. Currently, different therapeutic strategies have been implemented to attenuate IRI, though none have been considered definitive [8,9,10]. For example, ischemic pre-conditioning, which consists of brief and repetitive episodes of IRI before the induction of sustained organ ischemia, has been found to be effective for a number of operative settings where ischemia can be tightly controlled, such as transplantation, coronary bypass grafting, and elective major vascular procedures [11,12]. Ischemic post-conditioning is instead defined as the rapid and sequential intermittent interruption of blood supplies in the early stages of organ reperfusion. This therapeutic approach is usually applied when the ischemic damage cannot be predicted [13,14]. Finally, together with what is cited above, pharmacological treatments, despite their encouraging results in animal models of IRI, have not provided the expected results in a large number of clinical trials [8,15,16,17,18]. Consequently, it seems clear that new potential therapeutic approaches need to be identified and tested in order to reduce IRI side effects in various clinical settings.

## 2. Mesenchymal Stem/Stromal Cell (MSC)-Based Therapy as a New Strategy for Treating IRI

Scientific evidence has revealed that both cellular and acellular therapies based on the use of mesenchymal stem/stromal cells (MSCs), represent promising approaches to mitigate IRI-related pathological processes [19,20,21,22]. MSCs have been found in different tissues, including umbilical cord (UC-MSCs) [23], bone marrow (BM-MSCs) [24], adipose tissue (Ad-MSCs) [25], and placenta (AM-MSCs) [26], where they participate in the maintenance of stem cell niches and tissue homeostasis [27,28]. Several studies have demonstrated that MSCs possess therapeutic properties arising from their ability to secrete a plethora of functional factors involved in both immune regulation and tissue repair [29,30,31,32,33]. In particular, MSCs can secrete growth factors, cytokines, chemokines, and extracellular vesicles (EVs), such as exosomes (EXOs), which confer on MSCs paracrine therapeutic capabilities that can lead to, e.g., immune modulation, tissue injury amelioration, and reduction in fibrosis [32,34,35,36,37,38]. Oxidative stress is strongly correlated with cellular injury, and involved in the onset of several pathologies, including IRI [39,40]. Growing evidence supports the hypothesis that MSCs exert antioxidant properties in several pathological processes, which may explain MSCs’ cytoprotective properties in IRI experimental models [41,42]. These cells have been extensively tested in different areas of therapeutic application, including organ transplantation [43,44], where the beneficial strategies are aimed at reducing IRI and acute inflammatory responses (Table 1). Indeed, in many experimental models, including organ transplant studies, it has been found that infusion of MSCs promotes regeneration and prolonged recipient survival by reducing IRI and acute inflammation (Figure 1) [45,46,47,48,49,50,51].

In the future, the routine use of allogeneic MSCs may well present some hurdles to be overcome. In fact, MSCs can be found in the parenchyma of different tissues after intravascular administration, leading to the risk of tumor formation and/or the possibility of immune rejection for allogenic cells. Consequently, in order to obtain the therapeutic effects of MSCs without using the cells, MSC-derived products such as conditioned medium (CM) and EXOs have been investigated. For example, various in vivo studies of organ injury have highlighted the efficacy of CM derived from MSC cultures [52,53], supporting the emerging consensus that MSCs secrete bioactive factors that can mediate beneficial therapeutic effects through a paracrine mechanism. Indeed, individual components of the MSC secretome have been involved in a number of essential cellular processes, including angiogenesis, immunomodulation, wound healing, and tissue repair [32,54]. MSC-derived CM also contains functional factors that have shown therapeutic efficacy against ischemia reperfusion injury. In several IRI experimental models, it has been demonstrated that the whole secretome or EXOs alone were able to alleviate IRI side effects by attenuating inflammation and apoptosis, and improving tissue regeneration (Table 1).

**Table 1 biomedicines-11-00689-t001:** Summary of in vitro and in vivo studies reporting the use of MSCs and/or their products in preventing ischemia/reperfusion injury (IRI).

Use of Cells or Their Products	Study Model	Effects Due to MSC Treatment	References
AMSC-derived CM	In vitro model of human lung IRI	Attenuation of IRI effects by improving the efficacy of in vitro EVLP	[20]
AMSC-derived CM	In vitro model of hepatic IRI	Inhibition of activation of inflammatory macrophages and apoptosis in hepatocytes	[22]
UC-MSCs	Rat lung IRI	Reduction in oxidative stress damage and inflammation	[41]
BM-MSC-derived EVs	Mouse intestinal IRI	Mitigation of intestinal pathological injury, reduction in intestinal cell apoptosis and oxidative stress	[42]
UC-MSCs	Swine lung IRI	Attenuation of IRI by improving the efficacy of EVLP	[49]
BM-MSCs	Mouse lung IRI	Protection against cold IRI in lung transplants	[55]
AdMSCs	Rat lung IRI	Attenuation of inflammation and oxidative stress	[56]
BM-MSCs	Rat lung IRI	Reduction in both pulmonary edema and pro-inflammatory factors, and increase in anti-inflammatory factors	[57]
AdMSCs	Rat lung IRI	Attenuation of lung damage after IRI	[58]
BM-MSCs	Rat lung IRI	Attenuation of lung pathologic injury	[59]
BM-MSCs	Human lung IRI and EVLP	Decreased cold ischemic injury	[60]
BM-MSCs	EVLP in human lungs rejected for transplantation	Improvement in alveolar fluid clearance and reduction in both acute IRI and fibrotic responses	[61]
MSC-derived EVs	Rat lung IRI and EVLP	Improved tissue integrity and metabolism	[62]
UC-MSCs and UC-MSC-derived EVs	Mouse lung IRI	Attenuation of lung dysfunction and injury by improving the efficacy of EVLP	[63]
BM-MSC-derived CM	Rat lung IRI	Protection against lung IRI	[64]
BM-MSCs	Mouse model of kidney IRI	Induction of M1 to M2 transition and tissue regeneration	[65]
BM-MSCs	Human renal allograft model with exsanguinous metabolic support	Reduction in ischemic damage and inflammatory cytokines	[66]
BM-MSCs	Human renal allograft model with normothermic machine perfusion	Reduction in ischemia reperfusion injury	[67]
BM-MSCs	Rat intestinal IRI	Reduction in inflammatory response and intestinal ischemic damage	[68]
BM-MSCs	Rat intestinal IRI	Reduction in intestinal ischemic damage	[69]
BM-MSCs	Swine myocardial infarction model	Attenuation of contractile dysfunction and pathologic thinning	[70]
AMSCs	Swine myocardial infarction model	Reduction in histological and functional impairment of myocardium	[71]
BM-MSC-derived EVs	Rat myocardial infarction model	Improvement of blood flow recovery, reduction in infarct size and preservation of cardiac systolic and diastolic performance	[72]
MSC-derived CM	Rat cardiac allograft model with hypothermic perfusion	Improvement of cardiac function and reduction in pro-inflammatory cytokines	[73]
BM-MSC-derived CM and EVs	Hypothermic perfusion of mouse donor heart	Attenuation of ischemia-induced myocardial damage in donor heart and improvement of heart function after transplantation	[74]
BM-MSC-derived CM	Rat cardiac allograft model	Improvement of cardiac post-operatory functions and reduction in pro-inflammatory cytokines	[75]
BM-MSCs	Mouse model of embolic middle cerebral artery occlusion	Improvement in functional recovery	[76]
BM-MSCs	Rat model of embolic middle cerebral artery occlusion	Improvement of function, induction of angiogenesis, and reduction in apoptosis	[77]
BM-MSCs	Rat model of hepatic IRI	Protection from the progression of the damage and reduction in neutrophil infiltration	[78]
AdMSCs	Rat model of hepatic IRI and hepatectomy	Inhibition of hepatic apoptosis and improvement of tissue regeneration	[79]
AdMSCs	Rat model of hepatic IRI	Inhibition of inflammasome activation	[80]
AdMSC-derived CM	Swine model of hepatic IRI and hepatectomy	Attenuation of hepatic IRI and hepatectomy-induced liver damage	[81]
MSC-derived EVs	Mouse model of hepatic IRI	Attenuation of liver damage and improvement in liver regeneration after IRI	[82]
BM-MSC-derived EVs	Mouse model of hepatic IRI	Attenuation of liver damage and inflammatory responses	[83]
UC-MSC-derived EVs	Mouse model of hepatic IRI	Reduction in liver IRI by reducing apoptosis and inflammation	[84]
UC-MSC	Rat model of hepatic IRI	Attenuation of injury by inhibiting inflammation, neutrophil infiltration, and apoptosis	[85]

EVLP: ex vivo lung perfusion; EVs: extracellular vesicles; CM: conditioned medium; MSCs: mesenchymal stem cells; BM-MSCs: bone marrow-derived MSCs; AMSCs: amnion-derived MSCs; UC-MSCs: umbilical cord-derived MSCs; AdMSCs: adipose-derived MSCs; IRI: ischemia-reperfusion injury.

Therefore, the therapeutic potential of both MSCs and MSC-derived products, are of particular interest as a strategy for modulating injury due to ischemia/reperfusion in several IRI-related diseases. In this review, we summarize the major studies investigating the clinical efficacy of MSC-based therapy in different IRI experimental models.

### 2.1. MSCs and Ischemia/Reperfusion Injury in the Lung

Ischemia/reperfusion injury is the leading cause of postoperative dysfunction after lung transplantation (LTx) [86,87]. This pathological condition is characterized by ROS production (production of toxic molecules), increased inflammation, alveolar damage, and lung edema, with consequent injury to the lung parenchyma, which can result in both early primary graft dysfunction (PGD) and/or chronic lung allograft dysfunction (CLAD) [19,41,88,89,90,91]. Lungs are particularly susceptible to IRI which, together with infections, can contribute to lung rejection and post-transplant mortality [90,92]. Therefore, a reduction in IRI adverse effects is a crucial step to improving LTx.

Over the last decade, a number of reports have shown that MSCs and/or their products (CM and EVs) are able to stimulate tissue regeneration, inhibit immunological responses, and block ROS production [32,35,93,94,95,96,97,98]. They have been shown capable of decreasing inflammation and IRI in both in vitro and in vivo models, and those effects were mediated, at least in part, by the paracrine activity of MSCs (Figure 1) [99]. MSC therapeutic action has also been evaluated both in human lung diseases [19,100,101,102,103] and in different lung experimental models [19,55,97,104,105]. In those cases, MSCs and/or their products were capable of potentiating anti-microbial action, and mitigating both lung injury and inflammation, as found with IRI of LTx. Lin et al., in a rat model of lung injury, demonstrated that both xenogeneic and allogeneic MSCs were able to protect the lung against IRI by downregulating inflammation and oxidative stress [56]. Using the same model, Lu et al. showed that intravenous injection of BM-MSCs reduced both pulmonary edema and pro-inflammatory factors, while increasing anti-inflammatory factors [57]. In a rodent model, Sun et al. found that the autologous transplantation of Ad-MSCs was able to reduce lung IRI [58]. Moreover, Guillamat-Prats, Chen and colleagues demonstrated that the engraftment of allogenic MSCs reduced acute lung injury [59,106], while Chambers et al. showed that MSCs, infused via the peripheral vein twice weekly for 2 weeks, also decreased CLAD in human lung-transplant recipients [107].

Recently, it has been postulated that the use of normothermic ex vivo lung perfusion (EVLP) may be helpful in mitigating ischemic injury to the lung [108,109]. During EVLP, the organ is placed in a device, providing ventilation and perfusion for 4–6 h, and allowing the evaluation and recovery of compromised donor lungs [110,111]. Interestingly, for the duration of EVLP, while the estimation of donor lung function can be made, different treatments can also be used to further reduce IRI [15,18]. In this regard, MSC-based treatments (MSCs or MSC-derived products) have been tested to improve EVLP. The promising results obtained from those studies have led MSCs to be considered good candidates for integration with EVLP, aiming at improvement in LTx. Several reports have suggested that MSC-based treatment administered during EVLP is associated with a decrease in inflammation and ischemic injury of human donor lungs [49,60,112]. McAuley et al. showed that when MSCs were added to the EVLP perfusate, they can exercise the ability to restore alveolar fluid clearance, and reduce both acute IRI and fibrotic responses in human lungs rejected for transplantation [61]. Moreover, Pacienza et al., in a rat preservation model of lung injury, found that MSC treatment during EVLP was able to reduce oxidative damage and IRI [41]. Promising results have also been found by using CM or EVs obtained from MSC cultures. It has been shown that MSC-derived EVs were able to reduce inflammation, and enhance tissue regeneration in acute lung injury models [62,63,113]. In a mouse model of lung IRI, treatment with EVs derived from UC-MSCs improved the efficacy of EVLP, and attenuated lung dysfunction by decreasing pro-inflammatory cytokines, neutrophil infiltration, and edema [63]. Moreover, EVs derived from BM-MSCs, when administered during EVLP, were able to increase the alveolar fluid clearance in human donor lungs rejected for transplantation [114]. Similarly to EVs, MSC-derived CM has shown beneficial effects on different experimental models of lung diseases [20,64,115,116,117]. For example, BM-MSC-derived CM, when administered during EVLP, was able to increase alveolar fluid clearance in human lung injured by the E. coli endotoxin [117]. The use of MSC-derived products, rather than the direct use of cells, can avoid all risks associated with live-cell transplants and, therefore, represents an emerging and promising approach to treating IRI in the field of LTx.

The use of MSCs and/or their products is a promising approach to reducing IRI in several lung pathological conditions; therefore, the integration of MSC-based treatment with EVLP in the context of LTx seems to be the best way to improve LTx outcomes.

### 2.2. MSCs and Ischemia/Reperfusion Injury in the Pancreas

The transplantation of the whole pancreas or pancreatic islets is considered an effective treatment for restoring glycemia in specific patients with unstable type 1 diabetes mellitus (T1D). However, in the case of the whole pancreas, similarly to other solid organs, IRI pathological processes often lead to complications which occur after transplantation, resulting in poor transplant success [118]. On the other hand, the clinical outcome also needs to be improved for islet transplantation because a variable degree and length of insulin independency have been observed [119]. A number of aspects can affect clinical outcomes for those transplants and, among these, the quality of donor organ is often considered a key factor [120,121,122]. Pancreas or pancreatic islet transplantation techniques are characterized by different phases, including organ procurement, preservation, and islet isolation, during which IRI processes occur that affect organ/islet function and survival. In this case, IRI not only impacts on transplant outcomes, but also reduces the number of pancreas/islets suitable for transplantation.

To address those issues, a more effective preservation before transplantation represents a promising strategy for improving the quality of the pancreas/islets, and to achieving complete and long-term insulin independence [123,124,125,126]. For the pancreas, the most common organ preservation solution used is the University of Wisconsin (UW) solution, which, however, does not prevent the deleterious effects of IRI [127,128,129]. Recently, in a T1D mouse model, Nishime et al. demonstrated that UW solution, supplemented with the antioxidant and cytoprotective agent AP39, mitigates cold IRI and promotes higher islet yields before and after purification. In this study, graft pancreatitis was prevented, and the outcome of islet transplantation was improved [130].

Across the last decade, it has been shown that the treatment with MSCs was able to induce islet regeneration, increase islet function, and inhibit immune system reactions in various animal models of T1D [131]. Kasahara et al. found that AdMSCs-CM administered through a preservation solution was able to restore rat islets to the condition they were in before transport, culture, and transplantation [132]. Similar results were found by Teratani et al., in which the MSC-derived CM added to the UW solution was able to preserve porcine islets during both transportation and cultures [133].

In the early phase after islet transplantation, the lack of vasculature and the hypoxic environment contribute to islet acute injury, mediated by ischemia [134]. In addition, after transplantation, graft failure continues to occur because of immunological reactions [135]. MSCs have the ability to both stimulate angiogenesis and attenuate inflammation [32], and are thus considered a useful therapeutic tool for alleviating IRI and improving islet transplantation outcomes. MSCs can be utilized in the early phase of the transplant to suppress immune-mediated rejection. Moreover, MSCs can provide a favorable environment for improving islet engraftment and stimulating their regeneration [136]. In a mouse model of allogenic islet transplantation, it has been shown that MSCs were able to prevent islet allograft rejection, leading to long-term normoglycemia. Ding et al. demonstrated that those effects were due to the MSCs’ suppressive activity on the immune system, which reduced hypersensitivity responses to allogeneic antigens, and allowed the survival of allogeneic islet grafts [137]. In a rat model of allogenic islet transplantation, Longoni et al. observed the ability of both syngeneic and allogeneic MSCs to prevent acute rejection and prolong graft function. In this study, the efficacy of MSCs was related to a reduction in inflammation, and this effect was independent of the administration route [138]. Li et al., in a diabetic mouse model, investigated the mechanisms by which the co-transplantation of MSCs and allogenic islets alleviated allograft rejection. They showed that MSCs exerted immunosuppressive effects through inhibition of both T lymphocytes and the activation of dendritic cells, resulting in the survival of transplanted pancreatic islets [139]. Promising results have also been obtained in non-human primates. Berman and colleagues showed that the intra-portal or intra-venous co-transplantation of MSCs and pancreatic islets significantly enhanced islet engraftment and function, and further infusions of MSCs were also able, in some cases, to avoid rejection and maintain islet function [140].

Therefore, thanks to their beneficial properties (anti-oxidant, proangiogenic, and immunomodulatory properties), MSC-based therapies can prevent IRI pathological processes during the transplantation of either the whole pancreas or pancreatic islets. Moreover, addition of MSCs or their secreted products to preservation solution during organ preservation or isolation of islets could conceivably offer a novel pharmacological approach to improving the outcome of pancreas/islet transplantation.

### 2.3. MSCs and Ischemia/Reperfusion Injury in the Kidney

Currently, renal transplantation is the principal treatment option for patients with end-stage kidney disease (ESKD), and IRI is an inevitable event during renal transplantation, with a significant impact on the function of transplanted kidneys [141]. IRI is the main cause of acute kidney injury (AKI) in surgery, and is characterized by endothelial cell activation, and leukocyte recruitment and infiltration, as well as renal cell necrosis and apoptosis [142]. In one study, it was found that treatment with MSC-derived EVs can ameliorate kidney fibrosis in a cisplatinum-induced AKI mouse model by reducing the levels of ROS and of pro-apoptotic molecules, such as 8-hydroxy-2-deoxyguanosine (8-OHdG), malonaldehyde (MDA), Bax, and caspase-3 [143]. The regenerative and anti-inflammatory properties of MSCs have been explored in a large number of animal injury models, explicating how these cells promote endothelial repair [144,145,146,147,148]. In several phase I clinical trials for kidney disease and transplantation, it has been found that the effects of MSCs on endothelial cells might lead to an improved kidney transplantation outcome [149,150,151,152,153]. Among the factors involved in the AKI pathogenesis, inflammatory response runs through the entire process of IRI-derived AKI, in which innate immunity, principally through macrophages, has a pivotal role in both injury and repair processes [154,155]. Macrophages can differentiate in two different phenotypes: M1, which produces pro-inflammatory molecules, and M2, which can inhibit inflammation and promote the repair of injured tissue [156]. Hence, the M1/M2 ratio can regulate the progress of AKI toward chronic kidney disease (CKD) or, conversely, kidney repair [157]. In the early stage of IRI, M1 macrophages are the prevalent inflammatory cells in kidney tissue, while M2 macrophages appear in later stages. Thus, to alleviate kidney injury, it is necessary to eliminate pro-inflammatory M1 macrophages before IRI [158]. The transition of macrophages from M1 to M2 phenotype induces the production of growth factors, such as platelet-derived growth factor (PDGF), transforming growth factor beta 1 (TGF-β1), vascular endothelial growth factor A (VEGF-A), and insulin-like growth factor I (IGF-1), which promote the regeneration and repair of renal tubular epithelial cells [157,159,160]. In a recent study using a mouse model of kidney IRI, it was demonstrated that EXOs from indoleamine 2,3-dioxygenase-(IDO)-overexpressing BM-MSCs accelerates tissue regeneration upon IRI by inducing the M1/M2 transition [65]. To date, most of the research and clinical trials have been focused mainly on MSC therapy after kidney transplantation, but not prior to it [152]. The systemic administration of MSCs to transplant recipients is safe; however, it has been shown that the cells never reach the kidney, mostly because intravenously infused MSCs are largely trapped in the lungs [161,162,163]. By contrast, the administration of MSCs to donor kidneys in an ex vivo isolated organ-perfusion system will deliver cells directly to the injured organ. With this purpose in mind, normothermic machine perfusion (NMP) of donor organs has been recognized as an opportunity to maintain organ viability and allow therapeutic interventions prior to transplantation [164,165,166]. Two different studies of MSC renal infusions during NMP on discarded human kidneys reported beneficial effects of MSCs. In one case, the potentiation of renal regeneration by the increased synthesis of adenosine triphosphate (ATP), reduction in inflammatory response, increased synthesis of growth factor, and the normalization of the cytoskeleton and mitosis was found [66]. In the other case, a significant reduction in IRI through the improvement of clinically relevant parameters, such as urine output and micro-vascular perfusion, and injury biomarkers, such as the downregulation of interleukin (IL)-1β, upregulation of IL-10 and IDO, as well as decreased neutrophil recruitment were reported [67]. In conclusion, the use of MSCs or their derivatives represents a promising approach to treating kidney IRI and/or improving kidney transplantation outcomes.

### 2.4. MSCs and Ischemia/Reperfusion Injury in the Gut

Intestinal IRI is a pathological process characterized by local vasoconstriction, thrombosis, mitochondrial damage, inflammatory response, cellular damage, and cell death [167], resulting in an impaired intestinal mucosal barrier function [168], associated with severe clinical conditions such as extracorporeal circulation, mesenteric artery thrombosis, strangulated ileus, trauma, abdominal aortic aneurysm surgery, and intestinal transplantation [169,170].

The visceral inflammation caused by intestinal IRI can alter the epithelial barrier morphology and function, allowing bacterial translocation [171], and their products as pathogen-associated molecular patterns (PAMPs), from the lumen into the lamina propria [172], thus driving inflammation [173]. This process eventually results in endotoxemia, the release of multiple pro-inflammatory cytokines [174], systemic inflammatory response syndrome (SIRS) [50], and even multi-organ failure and death [175,176,177,178]. Therefore, in patients with critical illness, the development of effective therapies and discovery of novel agents capable of ameliorating intestinal IRI are crucial for reducing mortality.

The potential of MSC therapy has been widely discussed for the treatment of end-stage organ ischemia. Even in intestinal IRI, these cells boost functional recovery and limit inflammation, while the exact mechanisms have not yet been defined (Figure 1). It has been postulated that MSCs may play a protective role against intestinal IRI via exogenous migration into the damaged intestinal tissue, and differentiation into intestinal epithelial cells (IECs), to enhance the integrity of the gut barrier or through the exogenous release of paracrine and/or endocrine cytokines with anti-inflammatory, anti-apoptotic, and pro-angiogenetic characteristics [50], principally mediated by EXOs [179]. As for the first mechanism of action, there are few studies demonstrating the cellular differentiation of MSCs within the damaged bowel as the primary mechanism by which MSCs attenuate intestinal IRI [180,181]. Most evidence supports the hypothesis that paracrine mechanisms drive the therapeutic efficacy of MSCs within damaged tissue [182,183]. One of the most promising molecular targets downregulated by MSC therapy is NF-kB, known to activate the expression of several genes expressed during intestinal IRI, and involved in the inflammatory response process [184], including TNF-α [185], IL-1β [186], IL-6, and ICAM [187]. These molecules drive complementary activation and subsequent production of inflammatory mediators, including IL-8, IL-17, and IL-18 [188]. The role of MSC immunomodulation in intestinal IRI acts on one side by decreasing production of inflammatory mediators, such as TNF-α [68], IL-1β [189], and IFN-γ [190], and by increasing anti-inflammatory cytokine production, via monocytes’ stimulation, such as IL-10 [191]. As an alternative mechanism of action, it has been demonstrated that BM-MSCs are able to attenuate intestinal IRI by reducing tight junction disruption and ZO-1 downregulation, thereby restoring the intestinal mucosal barrier, likely by regulating the levels of TNF-α [69,192].

Several studies have found that MSCs upregulate expression of numerous growth factors, such as VEGF, FGF2, and TGF-β by a p38 MAPK-dependent mechanism, resulting in enhanced tissue restoration [193,194]. Among the multiple roles played by MSCs, there is increasing evidence that the efficacy of MSCs resides in the EXOs released, carriers of a discrete set of proteins, and coding and non-coding RNA. In vivo experiments have shown that MSC-derived exosomal miR-34a/c-5p, and miR-29b-3p improved the intestinal barrier, thus alleviating intestinal IRI via the Snail/claudin pathway [195]. Recently, it was reported that BM-MSC EXOs can alleviate intestinal IRI through the PTEN/Akt/Nrf2 pathway by targeting miR-144-3p [42].

### 2.5. MSCs and Ischemia/Reperfusion Injury in the Heart

Ischemic heart diseases are the leading cause of mortality worldwide, with more than 2 million deaths in 2019, 16% of the world’s total (https://www.who.int/, accessed on 11 December 2022). Transplantation is a vital option for patients with challenging heart failure, despite the limitations imposed by organ availability. The shortage in organ supply obviously depends on the number of potential and actual donors, but also in large part on organ preservation. In fact, organ deterioration after donor explant is a key issue in this life-saving procedure. Perfusion has been adopted for prolonging organ storage, providing oxygen and nutrients, and maintaining the conditions for organ preservation. Though perfusion techniques have improved since the first heart transplant, in 1967 [196], organ transport for location and matching constitutes a very demanding problem in the prevention of damage caused by ischemia and hypoxia. Temperature lowering can reduce the cell death process with the release of autolytic enzymes, without altering cell metabolism; however, prolonged ischemia and hypothermia can induce oxidative processes [197], swelling [198], and acidosis [199]. In order to reduce these effects, several protective formulations have been developed [200]. Currently, another pioneering strategy has been considered together with cold and chemical preparations, and based on the protective action of MSCs on various organs and, in particular, heart injuries [201]. Though not less than two decades ago MSCs were viewed as playing a directly plastic role in heart remodeling after injury [70,202], further experiments have demonstrated that MSCs act mostly via paracrine effects (Figure 1). The first experiments revealed MSCs’ effects in in vivo animal models after MI, identifying cardiomyocyte transdifferentiation, and the integration into cardiac tissues [71,203]. However, further experiments have highlighted a limited amount of transdifferentiated MSC-derived cardiomyocytes [203,204]. This opened an important debate on the integration of MSCs into cardiac tissues after injection, and subsequent cardiomyocyte transdifferentiation [205]. Further evidence has demonstrated that the therapeutic effects of MSCs on the heart are based mostly on factors released, since their action is exerted in a paracrine mode by the secretome [206]. For example, it has been shown that MSC-derived EVs promote angiogenesis, an effect that can be ascribed principally to the miRNA content in EVs [207]. MiRNAs have been involved at various levels in cardiac repair or cardiomyocyte proliferation/aging [208,209] by modulating pathways implicated in the inhibition of fibrosis after MI in repair processes. MSC-derived EVs can promote cardiac protection, given that this treatment shows a positive effect in rat models on the MI heart by reducing scar-size formation, and stimulating angiogenesis [72]. Proteomic analysis of MSC EVs has revealed the presence of growth factor receptors such as PDGFR, cytokines, signaling, and adhesion molecules [210], which account for the induction of angiogenesis. The effect of MSCs in improving hypothermic perfusion in ex vivo organ preservation was recently tested in allograft in vivo models [73,74,75]. The authors here found that the MSC secretome considerably improved post-operatory heart functions in rat allograft experiments [75]. Moreover, they showed that hypoxic MSC-CM have cardioprotective effects by significantly reducing apoptotic indexes (e.g., TUNNEL, DNA breaks). At the same time, pro-inflammatory cytokines have been reduced by MSC-CM treatment, with a stronger lowering of IL-6 and TNF-α in the hypoxic MSC-CM compared to normoxic MSC-CM. Similarly, another group evaluated MSC-CM in heart transplantation after extended organ storage in 15-month-old rats, which is approximately comparable to a 40-year-old human [73]. Myocardial evaluation of 120 genes involved in apoptosis, oxidative stress, and inflammatory response showed a significant modification in gene expression signature by MSC-CM treatment. This was associated with an improvement in the cardiac functions, such as a reduced re-beating time after transplant, and an enhancement of the LV systolic and diastolic functions, with an increase in the pressure rate [73]. Likewise, comparable results have been obtained in mouse models, where the MSC secretome determined cardiac protection against IRI after prolonged organ preservation [74]. MSC-CM substantially reduced the presence of pro-inflammatory cytokines (TNF-α, IL-1β, and IL-6) in allotransplanted hearts. Moreover, the authors found the presence of miR-199a-3p in MSC EVs, which showed a cardioprotective effect in IRI [211]. miR-199a-3p expression in donor hearts is strongly reduced after prolonged cold preservation. Thus, MSC EXOs containing miR-199a-3p can restore myocardial levels in experimental allograft models. The opportunity to modulate the release of miRNAs by engineering EXOs is an additional option, together with a MSC-based therapeutic approach [212]. The MSC secretome may turn out to be an efficient cell-free therapeutic option for reducing organ damage, and mitigating the problems experienced by long and slow-moving waiting lists for this life-saving approach.

### 2.6. MSCs and Ischemia/Reperfusion Injury in the Brain

Cerebral IRI is a common event in ischemic stroke, induced by the impaired blood supply to the brain for a short period of time, followed by its restoration [213]. The impaired blood supply is usually caused by a blood pressure perturbation, and induced by the presence of a thrombus or embolus in vessels. Depending on the site, size, and duration of cerebral ischemia, it can cause permanent and irreversible brain tissue damage, including, in the worst-case scenario, neuronal cell death and cerebral infarctions [214]. In recent years, the literature has extensively described the main mechanisms involved in IRI. Among the most studied are oxidative stress, leukocyte infiltration, mitochondrial mechanisms, activation and aggregation of platelets, and the prompting of protein complement. Notably, a substantial number of studies supports the hypothesis that inflammation is a key factor in initiating the process at the base of the pathogenesis of cerebral IRI, which culminates in severe neurological dysfunction, such as blood–brain barrier (BBB) disruption, and edema or hemorrhagic transformation [213,215]. The disease is characterized by a robust activation and release of cytokines, chemokines, adhesion molecules, and proteolytic enzymes, which exacerbate tissue damage [216]. The current novel therapeutic approaches aim to modulate the neuroinflammatory response in order to reduce the inflammatory mediators involved in tissue damage [214,216]. Recent studies have demonstrated the strong immunomodulatory capacity of MSCs in the field of tissue regeneration. It has been found that they are able to inhibit tissue inflammation by directly or indirectly targeting the immune response, and promote tissue repair by the stimulation of endogenous cell function and revascularization of the damaged tissue [32,217]. MSCs can also inhibit the pro-inflammatory T-cell response, leading to a reduction in IFNγ, thus promoting an anti-inflammatory environment, which leads, in turn, to an increase in IL-10 production that induces the protective activity of Tregs [202,218,219]. The current data in the literature indicate that the administration of MSCs intracranially (intrastriatal or intracerebroventricular) or intravascularly (intra-arterial or intravenous) improves the restoration of tissue damage after ischemia both in mouse and rat models. The authors confirm that during cerebral stroke, the transplanted MSCs migrate to the damaged brain tissue site, inhibiting apoptosis and stimulating the expression of a series of growth factors, including brain-derived neurotrophic factor (BDNF), nerve growth factor (NGF), basic fibroblast growth factor (bFGF), IGF, HGF, VEGF, and angiogenic and stem cell factors, which synergistically prompt the recovery of the neuronal functions [76,220,221]. The recovery function carried out by MSCs also occurs through the induction of angiogenesis, the reduction in apoptosis, the restoration of synapses and dendrites, and the promotion of axonal regeneration and differentiation of autologous neuronal stem cells [77]. Recently, in vitro studies on astrocytes and neurons undergoing ischemic damage nicely demonstrated that MSC-EVs strongly protect the cells from intracellular Ca^2+^ accumulation, a biological effect of ischemia, and cell death, thus promoting tissue regeneration. In particular, the authors showed that the MSC-EVs acted by suppressing PI3K-Akt pathway. Furthermore, proteomic analysis on MSC-EVs revealed that the protective effect might be related to the EV-protein content, including factors such as HGF, CXCL1, VEGF-A, and MIF, all associated with brain regeneration [222]. Despite the fact that at present the MSC-evoked neurorestorative effects on IRI are not clear, recent clinical reports carried out on humans have highlighted a series of beneficial effects obtained in the treatment of ischemic stroke. Qiao et al. demonstrated the safety and feasibility of the co-transplantation of neural stem/progenitor cells (NSPCs) and UC-MSCs in patients with ischemic stroke, observing an improvement in neurological function and daily life activities [223]. Consonant with this study, Jing et al., in a small pilot study, demonstrated the safety and efficacy of allogeneic stem cells in treating strokes in the middle cerebral artery territory [224]. More recently, a series of clinical trials (phase 1/2) have confirmed that autologous MSCs, if delivered intravenously, are able to reduce post-stroke IRI damage, inducing a reduction in infarct size and an improvement in the functional outcome of the patients [225,226].

### 2.7. MSCs and Ischemia/Reperfusion Injury in the Liver

IRI in the liver occurs frequently during surgery for intrahepatic lesions, and is considered an inevitable injury during organ transplantation [227,228,229]. This is due mainly to the increasing number of potential recipients and the subsequent necessity of extending the criteria of liver eligibility for transplantation [230]. Thus, clinical needs require the identification of therapies that can attenuate the damage upon organ resection. Stem-cell therapy is currently considered an important approach, and has garnered attention in bench to bedside for the treatment of different diseases [32,231]. In particular, MSCs, because of their immunomodulatory effects and their tissue regeneration potential, have been used in liver surgery, including IRI [78,232,233]. Over the last decade, several studies from different research groups have used MSCs or their cellular products to detect protection from IRI, as well as to identify the molecular mechanisms underlining this biological effect (Figure 1). In vivo infusion of MSCs in mouse and rat models of liver IRI have shown an important protection against the injury. Pan et al., in 2012, found that BM-MSC infusion in IRI-injured rat livers protected the animals from the progression of the damage by reducing serum biomarkers of injury (AST and ALT), and by inhibiting neutrophil infiltration [234]. Furthermore, other studies have arrived at similar results in rat models, in which administration of Ad-MSCs inhibited hepatic apoptosis and increased tissue regeneration [79,235]. Reduced neutrophil infiltration upon BM-MSC infusion was confirmed a few years later by Li et al., who found that BM-MSCs stimulate MAPK phosphorylation and CXCR2 downregulation, both at transcriptional and protein levels. Taken together, these findings were associated with a significant reduction in the number of CD11^+^/CD18^+^ cells [80]. Along with the latter, MSC infusion in vivo inhibits pro-inflammatory phenotype during early organ reperfusion by modulating the transcription and the protein release of several pro-inflammatory cytokines, including TNF-α, IL-1β, and IL-6. More importantly, two different groups assessed the important role of MSCs (from bone marrow or adipose tissue) in regulating NLRP3 inflammasome activation, leading to reduced IL-1β and IL-18 protein release [236,237]. Interestingly, by using different mouse models, Li et al. discovered that BM-MSCs reduced NLRP3 activation via the Hippo/Wnt signaling axis [237]. MSCs either from bone marrow or adipose tissue have been employed in liver IRI attenuation because of their capacity to activate autophagy and mitophagy [81,238]. Wang et al. recently found that a BM-MSC infusion activates autophagy, as shown by increased levels of LC3b, a typical marker of terminal autophagy [239]. Moreover, they demonstrated that activation of BM-MSC-mediated autophagy depends on heme-oxygenase-1 (HO-1) upregulation. Furthermore, a recent paper by Zheng’s group showed that UC-MSCs activate mitophagy, a mitochondrial-related type of autophagy, thus promoting mitochondrial stability and cell survival [240]. Along the same line, the authors further confirmed mitochondrial protection by increased superoxide dismutase 1 (SOD1) activation and mitochondrial ROS (mtROS) inhibition after in vivo injection of UC-MSCs [240]. Endoplasmic reticulum stress (ERS), responsible for protein misfolding and subsequent cellular apoptosis, has recently been shown to be involved in liver IRI progression. Interestingly, it was recently found that Ad-MSCs attenuate ERS by downregulating ERS-related genes (ATF6 and XBP1), and by inhibiting hepatic apoptosis [241]. Viewed together, these findings clearly demonstrate that in vivo infusion of MSCs ameliorates liver IRI by tackling different biological processes that are usually responsible for the progression of the damage. However, in the last few years, mounting evidence suggests that the biological function exerted by MSCs might depend on their biological products, including the whole secretome or, more specifically, the EVs. As important cell–cell communicators, they have been indicated as a new therapeutic approach that could eventually replace infusion of MSCs, thus avoiding the risk of tumorigenicity, pulmonary embolism, and alloimmune response [82,242]. Independent of the source, MSC secretome and EVs have been shown to significantly reduce IRI damage by reducing ALT and AST blood levels, and by accelerating tissue regeneration upon injury [83,243]. In particular, Anger et al. demonstrated that treatment with UC-MSC-derived EVs was able to induce a drastic reduction in pro-inflammatory proteins, including HMGB1 and IL-1β, and a strong reduction in the expression of ICAM-1, a well-known adhesion molecule expressed in endothelial sinusoidal cells, and involved in neutrophil recruitment and infiltration. These data are in line with the results from Yao’s group [244]. In this study, the authors further characterized EV content, thus proposing Mn-SOD as a target protein involved in reduced IRI-mediated oxidative stress [244]. Moreover, UC-MSC EVs have been shown to inhibit CD4^+^ T cell activation in liver IRI. In particular, through unbiased proteomic approaches, the authors demonstrated that UC-MSC EVs contain CCT2, a Ca^2+^ modulator protein, which in turn downregulates NFAT1, thus reducing CD154 and the subsequent pro-inflammatory role of CD4^+^ T cells [245]. MSC EVs are enriched in miRNA, which are responsible for post-transcriptional changes in the target cells. Interestingly, only a few studies have identified potential miRNAs within MSC EVs that can attenuate liver IRI damage. For example, mir-1246 contained in UC-MSC EVs seems to alleviate liver IRI by reducing apoptosis and inflammation of parenchymal cells, and by modulating T helper/Treg balance [84,85]. Here, the authors showed that miR-1246 activates Wnt signaling in the target cells during IRI by inhibiting GSK3β, thus sustaining proliferation and inhibiting inflammation and apoptosis [85]. The same group in back-to-back studies demonstrated that miR-1246 modulates the T helper/Treg ratio by inhibiting the IL6ST-gp130-STAT3 signaling axis in an in vitro model of hepatic IRI [84]. Across recent years, mounting evidence suggests that MSC pre-conditioning might be useful to specify secretome content [32]. However, very few studies have focused on this type of approach in the context of liver IRI. In 2018, Sun et al. demonstrated in vivo that the 3D culture of UC-MSCs generates spheroids that, when implanted in an IRI-damaged liver in rats, attenuates the injury by inhibiting inflammation, neutrophil infiltration, and parenchymal apoptosis [246]. Finally, our group recently showed that 3D and IFN-γ priming of human amnion-derived MSCs (AMSCs) are able to modulate IRI damage in an in vitro model of IRI carried out on primary macrophages and hepatocytes. We found that several bioactive factors, including HGF, IL-10, IL-1RA, and BDNF, selectively inhibit inflammation of primary macrophages and apoptosis of IRI hepatocytes in vitro [22].

## 3. Discussion

The main biological processes activated during IRI are shared among the different types of organs where the damage occurs. As described above, inflammation, ROS production, and apoptosis of the parenchymal cells are responsible for the start and progress of the injury, leading eventually to organ rejection. Thus, it does not seem unusual that proteins and factors released during the injury are common in all the damaged tissues. For these reasons, using MSCs for the treatment of multi-organ IRI is currently considered a valid approach to reducing the injury (Figure 1) [21]. The scientific literature is replete with interesting studies that report the resolution of IRI after MSC treatment [50,56,67,75,80,226,247]. Because of their properties, MSCs can modulate the damage at different levels. For example, they strongly reduce inflammation, thus inhibiting the release of several pro-inflammatory cytokines, which in turn regulate neutrophil infiltration in the damaged tissue [192,248,249]. In addition, MSCs have been shown to block ROS production, thus improving mitochondrial damage [192]. Altogether, the inhibition of such biological mechanisms eventually leads to the reduction in parenchymal cell death by preventing organ rejection. Currently, there are seven clinical trials registered on clinicaltrials.gov, aimed at investigating MSC therapeutic efficacy for the treatment of IRI for both heart and kidney. Moreover, MSCs have demonstrated significant therapeutic effects in numerous preclinical IRI models (Table 1) and in the context of human solid-organ transplantation. In particular, in different preclinical human studies in which organs from deceased donors were not suitable for transplantation, it has been shown that MSCs were able to reduce IRI side effects in both lung and kidney in the setting of mechanical organ perfusion [60,61,66,67]. However, despite the safety of MSC treatment having been widely demonstrated for different diseases in numerous patients [250], several studies have also reported adverse events and side effects associated with MSC therapy [251]. For instance, the long-term cell culture of MSCs can lead to genetic abnormalities with consequent tumorigenic effects. Moreover, different clinical trials reported fibrosis and thromboembolism as the most common adverse events of MSCs therapy [251]. All these problems may be avoided by exploiting the paracrine properties of MSCs. Interestingly, in the recent past it has been demonstrated that the MSC secretome accounts for most of the positive effects of MSC therapy for IRI. The MSC secretome, containing biological factors that include EVs enriched in proteins and miRNAs, has been shown to be strongly involved in the modulation of the IRI phenotype for several reasons. First, it contains immunomodulatory factors, as well as pro-angiogenic and tissue repair properties, which by themselves seem to account for the majority of the therapeutic effects [20,22,252,253,254]. Second, because of the MSC secretome requirement for resolution of IRI, its use has recently been proposed instead of MSC transplantation. This has prompted reflection, as several studies have demonstrated that MSCs, when intravenously transplanted, die before tissue homing, becoming trapped in the lungs, or because they do not survive the highly hypoxic environments generated by ischemic tissues [162,255,256,257]. MSC paracrine effects seem to be more efficient, thus suggesting that MSC secretome-based therapy might be beneficial for IRI treatment. In the last few years, mounting evidence has suggested that MSC priming may be the future of MSC-based therapies. Many recent studies, mostly in vitro, have shown that MSC pre-conditioning might specify the secretome content, thus allowing the selection of the type of factors within the CM according to the disease to be tackled. For instance, it has been shown that IFNγ or IL-17 MSC priming generates a secretome enriched in anti-inflammatory factors, setting the stage for its use in inflammatory diseases, including IRI [22,258]. In addition, if the resolution of the injury requires tissue regeneration or angiogenesis, 3D culture priming, and hypoxia pre-conditioning stimulate the generation of MSC secretomes enriched in molecules, growth factors, and EVs required for tissue repair or formation of new vessels [259,260,261]. Very few clinical studies have described strategies to attenuate IRI damage, though the in vitro results obtained in the recent past have paved the way for further development of this novel approach in preclinical studies.

## 4. Conclusions

IRI damage currently represents an important impairment in organ surgery, given the variety of biological processes involved, and the multiple cell types targeted during the injury. Inflammation, ROS production, and apoptosis are the main consequences of the damage, thus therapeutic interventions are required and new strategies are under investigation. MSC-based cell therapy seems to be one of the most promising, considering the immuno-modulating and tissue regeneration properties provided by this cell type. However, several issues need to be addressed, as MSCs’ injection still has adverse effects in the short term, while no data are yet available in the long term. New alternatives are in place, which implies MSC-cell-free therapies; however, the studies are still in their infancy and further research needs to be performed before validation in pre-clinical human subjects.

## Figures and Tables

**Figure 1 biomedicines-11-00689-f001:**
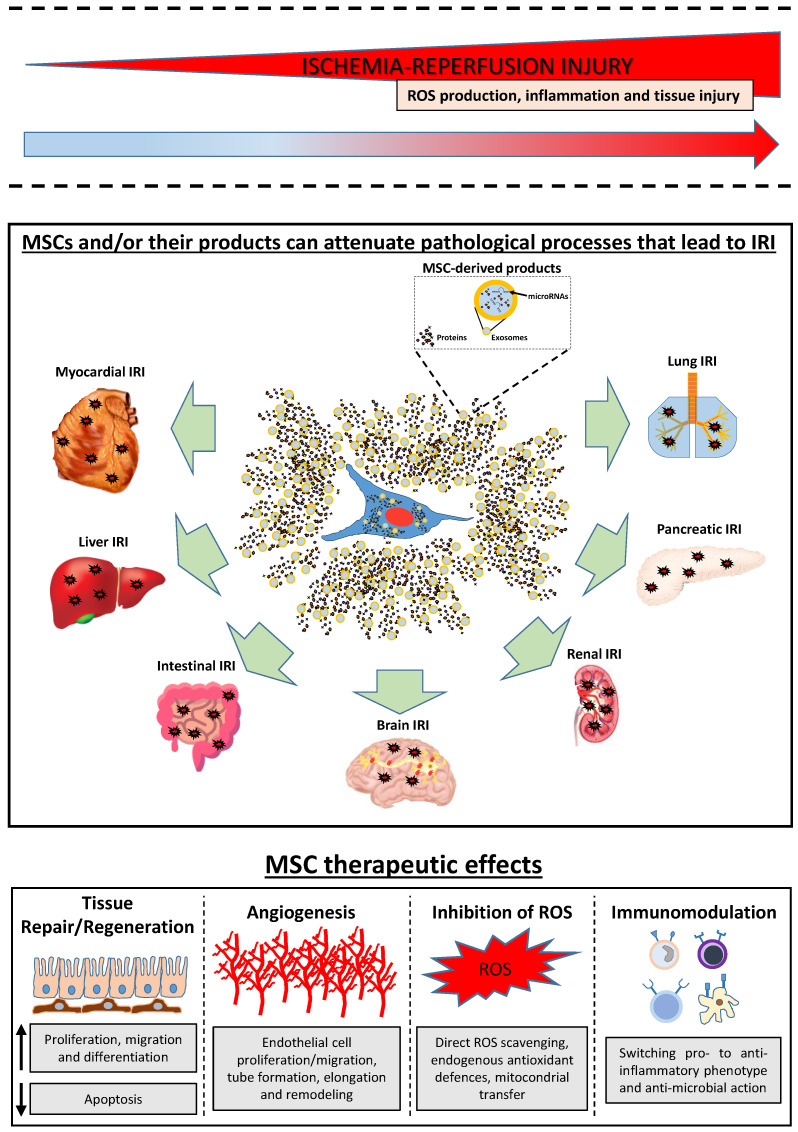
Schematic representation of MSC effects on IRI, with the main biological processes involved in the damage that are attenuated with MSC therapy.

## Data Availability

No new data were created or analyzed in this study. Data sharing is not applicable to this article.

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
