# Peer review of "Role of Mesenchymal Stem/Stromal Cells in Modulating Ischemia/Reperfusion Injury: Current State of the Art and Future Perspectives"

_biomedicines, 2023, doi:10.3390/biomedicines11030689_

Round 1

Reviewer 1 Report

1) Some conclusions of the literature review on the role of mesenchymal stem/ stromal cells in modulating ischemia/reperfusion injury should be also provided in the abstract;

2) The authors should highlight the novelty and the requirement of a new review in the light of existing data (a review from 2015: https://pubmed.ncbi.nlm.nih.gov/26258151/);

3) The authors reported in Table 1 in vitro and in vivo studies on the use of MSCs in preventing ischemia/reperfusion injury. Although some data were presented in main text on MSCs and heart ischemia/reperfusion injury prevention (including myocardial infarction), those data were not presented in the table. Thus, table 1 should be updated to include information on ischemic heart disease;

4) It would be useful to discuss and highlight specific MSCs that entered clinical trials on humans regarding ischemic/reperfusion injury prevention;

5) Ongoing preclinical as well as clinical studies on MSCs should be presented (e.g., in discussion section;

6) Some potential side effects and drawbacks of MSCs therapy should be discussed, in order to enhance further research on human subjects;

7) Conclusions should be reported as an individual chapter and should include the main findings of the literature review.

Author Response

1) Some conclusions of the literature review on the role of mesenchymal stem/ stromal cells in modulating ischemia/reperfusion injury should be also provided in the abstract

A: we thank the reviewer for the suggestion, we edited the abstract according the requests

2) The authors should highlight the novelty and the requirement of a new review in the light of existing data (a review from 2015: https://pubmed.ncbi.nlm.nih.gov/26258151/);

A: we are grateful with the reviewer for bringing up this aspect. We are aware of the review from Rowart et al, 2015. In this manuscript the authors provided an important update on Ischemia/Reperfusion Injury biology and the role of MSCs in modulating the damage in pre-clinical animal model of kidney, liver, heart and brain. Along with what cited before, with their related scientific updates, our review provides for the first time detailed information on the role of MSCs in other organs undergoing IRI, including lung, pancreas and gut.  In addition, it is important to point out that Ischemia/Reperfusion Injury is an important topic that in the last years brought up new knowledges related to new molecular mechanisms involved in acute IRI phase, as showed by the numerous references we included in the manuscript posterior to 2015. Furthermore, even the MSC therapy for IRI treatment further developed, as showed by tremendous increase of studies using the MSC conditioned medium or exosomes. Finally, new in vitro approaches are described which use pre-conditioned MSCs to further improve IRI outcome. Thus, we believe that the current manuscript provides novel information to the IRI/MSCs field given the rapid growth of this research topic in the last years.

3) The authors reported in Table 1 in vitro and in vivo studies on the use of MSCs in preventing ischemia/reperfusion injury. Although some data were presented in main text on MSCs and heart ischemia/reperfusion injury prevention (including myocardial infarction), those data were not presented in the table. Thus, table 1 should be updated to include information on ischemic heart disease

A: We thank the reviewer for the suggestion, we edited the table according to the request

4) It would be useful to discuss and highlight specific MSCs that entered clinical trials on humans regarding ischemic/reperfusion injury prevention;

A: We thank the reviewer for the suggestion. Currently, 7 clinical trials have been registered on ischemia/reperfusion injury and MSC treatment. This information has been added and described in the discussion of the current manuscript.

5) Ongoing preclinical as well as clinical studies on MSCs should be presented (e.g., in discussion section;

A: as for point 4, we have edited the discussion section by adding description and comments on some pre-clinical studies performed in human lungs and kidneys obtained from donors and that could not be used for organ transplant.

6) Some potential side effects and drawbacks of MSCs therapy should be discussed, in order to enhance further research on human subjects;

A: This is a very important aspect that need to be developed. In the discussion section of the manuscript we have commented on that providing the most important problems related to IRI infusion and/or injection, such as block of MSCs in the lungs and lack of survival in a hypoxic environment generated by ischemic tissues. We have added more information in this context which might easily bring the reader to the next part of the review which proposes the use of MSC secretome for further research on human subjects.

7) Conclusions should be reported as an individual chapter and should include the main findings of the literature review.

A: we agree with the reviewer and decided to split discussion and conclusion of the current manuscript.

Reviewer 2 Report

This is an interesting and relevant article. The authors have done a lot of work. Section 2.6. MSCs and ischemia/reperfusion injury in the brain should discuss the cytoprotective mechanisms of mesenchymal stromal cell-derived extracellular vesicles. https://pubmed.ncbi.nlm.nih.gov/36147480/

Author Response

We thank the reviewer for the insightful suggestion and added the citation in the paragraph as requested

Round 2

Reviewer 1 Report

Great! Congrats!